# RAE: A Neural Network Dimensionality Reduction Method for Nearest Neighbors Preservation in Vector Search

## Abstract

While high-dimensional embedding vectors are being increasingly employed in various tasks like Retrieval-Augmented Generation and Recommendation Systems, popular dimensionality reduction (DR) methods such as PCA and UMAP have rarely been adopted for accelerating the retrieval process due to their inability of preserving the nearest neighbor (NN) relationship among vectors. Empowered by neural networks' optimization capability and the bounding effect of Rayleigh quotient, we propose a Regularized Auto-Encoder (RAE) for k-NN preserving dimensionality reduction. RAE constrains the network parameter variation through regularization terms, adjusting singular values to control embedding magnitude changes during reduction, thus preserving k-NN relationships. We provide a rigorous mathematical analysis demonstrating that regularization establishes an upper bound on the norm distortion rate of transformed vectors, thereby offering provable guarantees for k-NN preservation. With modest training overhead, RAE achieves superior k-NN recall compared to existing DR approaches while maintaining fast retrieval efficiency.

## 1 Introduction

### 1.1 Background and Motivation

Vector embeddings have become the cornerstone of modern AI systems, enabling sophisticated semantic understanding across diverse domains including Information Retrieval (Zhu et al. (2023)), Recommendation Systems (Zhao et al. (2024)), and Retrieval-Augmented Generation (RAG) pipelines (Gao et al. (2023)). These embeddings, typically generated by pre-trained models such as BERT (Reimers & Gurevych (2019)), CLIP (Radford et al. (2021)), or LLMs (Achiam et al. (2023); Touvron et al. (2023)), encode complex data into high-dimensional vector spaces where semantic similarity corresponds to metric proximity. Vector databases have emerged as specialized infrastructure to efficiently store and retrieve these embeddings at scale (Wang et al. (2021); Douze et al. (2024)). However, the high dimensionality of these embeddings, commonly with hundreds or even thousands of dimensions, poses significant computational and storage challenges. The curse of dimensionality (Aggarwal et al. (2001)) manifests in multiple ways: (i) similarity computation scales linearly with dimension, making real-time retrieval computationally expensive; (ii) storage requirements grow proportionally, limiting scalability for billion-scale collections; and (iii) Nearest Neighbor search becomes difficult because distances tend to concentrate as dimensions increase, which motivated extensive Approximate Nearest Neighbor (ANN) research (Malkov & Yashunin (2018)). These challenges motivate the exploration of dimensionality reduction (DR) as a potential solution to accelerate retrieval while maintaining semantic fidelity.

Traditional DR techniques, including Principal Component Analysis (PCA) (Wold et al. (1987)), t-SNE (Maaten & Hinton (2008)), and UMAP (McInnes & Healy (2018)), have been extensively studied and applied across various domains. While these methods effectively compress high-dimensional data, they are fundamentally designed with different objectives: PCA maximizes variance preservation, t-SNE optimizes visualization of local structures, and UMAP maintains topological continuity (Van Der Maaten et al. (2009)). Critically, none of these methods explicitly optimizes for k-nearest neighbor (k-NN) preservation, which is the core requirement for retrieval tasks. Consequently, when

applied to embedding vectors, these methods often disrupt the neighborhood structure essential for accurate similarity search, making the reduced vectors unsuitable for direct use in retrieval applications.

Despite these limitations, the potential benefits of DR remain compelling. Reduced vectors enable faster distance computations, smaller index structures, and more efficient memory utilization, advantages that become increasingly important as embedding-based applications scale. This gap between efficiency gains and retrieval accuracy loss motivates our central research question: can we develop a dimensionality reduction method that explicitly preserves k-NN relationships while achieving substantial compression ratios?

## 1.2 Proposed Work

To address this challenge, we propose RAE (Regularized Auto-Encoder), a neural network-based dimensionality reduction framework specifically designed to preserve k-NN relationships after dimension reduction. Unlike traditional methods that optimize for other metrics, RAE directly targets the preservation of relative distances between vectors and their neighbors through a carefully crafted optimization objective.

Our approach leverages the architecture of encoder and decoder (Rumelhart et al. (1986)) combined with a novel regularization strategy to learn the reduced representation during the reconstruction process. The core idea is simple but effective: by introducing a regularization term with a controllable coefficient into the optimization objective, we guide the learning process to favor transformations that maintain neighborhood structures. This regularization mechanism enables RAE to learn representations that balance reconstruction loss with retrieval accuracy.

What distinguishes RAE from existing DR methods is its principled approach to k-NN preservation. Rather than relying on heuristics or indirect objectives, we establish a direct mathematical connection between our regularization strategy and neighborhood preservation quality. Through theoretical analysis, we prove that our method provides bounded guarantees on how well the reduced vectors maintain their original neighbor relationships. This theoretical grounding not only explains why RAE works well but also provides practical guidance for parameter adjustment in different application scenarios.

In practice, RAE offers an attractive combination of performance and efficiency. It achieves superior k-NN preservation while maintaining low computational time costs. The method is model-agnostic, working with embeddings from any source without requiring access to the original models. These properties make RAE a practical drop-in solution for accelerating vector retrieval systems across diverse applications, from semantic search to recommendation systems.

## 1.3 Contributions

This paper makes the following key contributions:

- We propose **RAE**, a regularized autoencoder framework for dimensionality reduction that achieves superior k-NN preservation accuracy through incorporating a carefully designed regularization term with a real-valued coefficient $\lambda$. The complete framework and loss function will be elaborated on in Section 3.2.

- We provide a rigorous mathematical analysis proving that the regularization establishes an upper bound on the norm distortion rate of transformed vectors, thereby offering provable guarantees for k-NN preservation. By leveraging the property of Rayleigh quotient, we demonstrate that the coefficient $\lambda$ directly controls this upper bound, which in turn ensures the preservation of the neighborhood structure. Theoretical foundations are demonstrated in Section 3.3.

- Extensive experiments across diverse datasets demonstrate that RAE consistently outperforms other dimensionality reduction baselines in k-NN preservation tasks. With reasonable pre-training time, RAE has comparable inference speed, supporting large-scale real-time computation. (Section 4)

## 2 RELATED WORK

### 2.1 CLASSICAL DIMENSIONALITY REDUCTION METHODS

Principal Component Analysis (PCA) (Pearson (1901); Abdi & Williams (2010)) remains the most widely adopted dimensionality reduction technique due to its simplicity and efficiency. PCA identifies orthogonal directions of maximum variance through eigenvalue decomposition, selecting the top k components for optimal data representation. Linear Discriminant Analysis (LDA) (Blei et al. (2003)) extends this framework by incorporating class labels to maximize inter-class separation. Random Projection (Achlioptas (2001); Vempala (2005)) offers a data-independent alternative that preserves pairwise distances with theoretical guarantees based on the Johnson-Lindenstrauss lemma (Frankl & Maehara (1988); Matoušek (2008)). While these linear methods are computationally efficient and theoretically well-understood, they optimize for global objectives rather than local neighborhood structure.

### 2.2 MANIFOLD LEARNING AND NONLINEAR METHODS

Manifold learning methods aim to discover low-dimensional structures in high-dimensional data by preserving local geometric properties. Isomap (Tenenbaum et al. (2000)) maintains geodesic distances on the data manifold, while Locally Linear Embedding (LLE) (Roweis & Saul (2000)) preserves local linear relationships. t-SNE (Maaten & Hinton (2008)) and its scalable variant UMAP (McInnes & Healy (2018)) have become popular for visualization, optimizing the preservation of local neighborhood probabilities through divergence minimization.

Despite their focus on local structure, these methods still do not focus on k-NN preservation tasks. They either require prohibitive computational costs for graph construction (Isomap, LLE) or sacrifice distance metrics for visualization clarity (t-SNE, UMAP). Furthermore, their transductive nature prevents efficient out-of-sample projection, making them impractical for real-time retrieval systems compared to our direct optimization approach for k-NN preservation.

### 2.3 DEEP LEARNING METHODS FOR DIMENSIONALITY REDUCTION

Modern deep neural networks inherently possess powerful encoding capabilities that enable dimensionality reduction as a byproduct of representation learning. Pre-trained models like BERT (Reimers & Gurevych (2019)) and GPT (Brown et al. (2020)) project text into fixed-dimensional embeddings, effectively performing implicit dimensionality reduction from sparse token spaces. Several methods have been proposed to explicitly compress these embeddings: BERT-whitening (Su et al. (2021)) applies whitening transformation to improve isotropy and reduce redundancy, while Matryoshka Representation Learning (MRL) (Kusupati et al. (2022)) trains models to produce nested representations where prefixes of the embedding vector form meaningful lower-dimensional representations.

However, these deep learning approaches suffer from fundamental misalignment with k-NN preservation objectives: they typically optimize for downstream classification or generation tasks rather than maintaining neighborhood structures, and require end-to-end training of the entire model with substantial computational resources.

## 3 METHODOLOGY

We propose a new dimensionality reduction method based on neural network called Regularized Auto-Encoder (RAE), which aims to preserve k-NN relationship between high-dimensional embeddings. While existing methods like PCA and UMAP, which maintain global maximum variance and local topological structure separately, RAE directly targets k-nearest neighbor preservation as its primary design criterion. RAE employs two single-layer fully-connected networks as encoder and decoder respectively, trained using reconstruction loss combined with regularization on parameter matrices. The trained encoder serves as the final dimensionality reduction mapping function.

The remainder of this section proceeds as follows. Section 3.1 formally defines the k-NN preserving problem and evaluation metrics. Section 3.2 presents the general RAE architecture and

our theoretically-motivated objective function combining reconstruction loss with Frobenius norm regularization. Section 3.3 specifically establishes our theoretical foundation by connecting k-NN preservation to norm-bounded transformations and deriving singular spectrum bound via Rayleigh quotient property.

## 3.1 PROBLEM FORMULATION

Let $\mathcal{X} = \{x_1, x_2, ..., x_N\} \subset \mathbb{R}^n$ be a collection of $N$ high-dimensional embedding vectors. We seek to learn a dimensionality reduction mapping $f : \mathbb{R}^n \rightarrow \mathbb{R}^m$ with $m < n$, which projects the original embeddings onto a lower-dimensional manifold $\mathcal{X}' = f(\mathcal{X}) = \{x'_1, x'_2, ..., x'_N\} \subset \mathbb{R}^m$.

Both the source space $(\mathbb{R}^n, d_{\mathcal{X}})$ and target space $(\mathbb{R}^m, d_{\mathcal{X}'})$ are endowed with distance functions satisfying the metric axioms: (i) non-negativity and identity of indiscernibles: $d(x, y) \geq 0$ with equality iff $x = y$; (ii) symmetry: $d(x, y) = d(y, x)$; and (iii) triangle inequality: $d(x, z) \leq d(x, y) + d(y, z)$ for all points $x, y, z$. Throughout this work, we focus on two ubiquitous metrics in vector retrieval systems: the Euclidean distance $d_E(x, y) = \|x - y\|_2$ and the cosine distance $d_C(x, y) = \frac{x^T y}{\|x\|_2 \|y\|_2}$, both of which constitute proper metric spaces.

**Definition 1** (k-NN Preservation Task). Given an anchor point $x_a \in \mathcal{X}$, let $\{x_i\}_{i=1}^k$ denote its k-nearest neighbors in the original space, characterized by:

$$d(x_a, x_i) \leq d(x_a, x_j), \quad \forall x_j \in \{x_j\}_{j=1}^{N-k-1} = \mathcal{X} \backslash (\{x_i\}_{i=1}^k \cup \{x_a\}) \tag{1}$$

A mapping $f$ is said to preserve the k-NN structure *as completely as possible* if, for all $x_j \in \{x_j\}_{j=1}^{N-k-1}$, *the more* $x_i \in \{x_i\}_{i=1}^k$ that satisfy the following relationship:

$$d(f(x_a), f(x_i)) \leq d(f(x_a), f(x_j)) \tag{2}$$

This constraint ensures that the relative proximity ordering induced by the k-nearest neighbors is preserved under the transformation $f$.

To formalize the notion of k-NN preservation quality, we introduce the following notation. For each point $x_a \in \mathcal{X}$, define:

$$\mathcal{N}_k^{\mathcal{X}}(a) = \{i : x_i \in \text{k-NN}(x_a) \text{ w.r.t. } d_{\mathcal{X}}\}$$

$$\mathcal{N}_k^{\mathcal{X}'}(a) = \{i : x'_i \in \text{k-NN}(x'_a) \text{ w.r.t. } d_{\mathcal{X}'}\}$$

where k-NN$(x)$ denotes the set of k-nearest neighbors of point $x$ under the respective metric.

**Definition 2** (k-NN Preservation Accuracy). The preservation rate at single anchor point $a$ is defined as:

$$P_a = \frac{|\mathcal{N}_k^{\mathcal{X}}(a) \cap \mathcal{N}_k^{\mathcal{X}'}(a)|}{k} \tag{3}$$

The overall k-NN preservation accuracy is given by the expectation over all points:

$$P_{\text{overall}} = \frac{1}{N} \sum_{a=1}^N P_a = \frac{1}{kN} \sum_{a=1}^N |\mathcal{N}_k^{\mathcal{X}}(a) \cap \mathcal{N}_k^{\mathcal{X}'}(a)| \tag{4}$$

**Optimization Objective.** Our goal is to identify the optimal transformation that maximizes k-NN preservation accuracy:

$$f^* = \arg \max_{f \in \mathcal{F}} P_{\text{overall}}(f) \tag{5}$$

where $\mathcal{F}$ denotes the feasible function space of mappings from $\mathbb{R}^n$ to $\mathbb{R}^m$. In subsequent sections, we demonstrate that restricting $\mathcal{F}$ to the class of norm-bounded linear transformations yields both theoretical guarantees and computational tractability.

## 3.2 RAE ARCHITECTURE

The RAE (Regularized Auto-Encoder) employs a simple yet effective linear architecture designed to achieve dimensionality reduction while preserving k-NN structure. The model consists of two linear transformations: an encoder $W_e \in \mathbb{R}^{m \times n}$ that maps from the original $n$-dimensional space to an $m$-dimensional latent space (where $m < n$), and a decoder $W_d \in \mathbb{R}^{n \times m}$ that reconstructs the original representation:

$$\hat{x} = W_d W_e x \tag{6}$$

where $x \in \mathbb{R}^n$ is the input vector and $\hat{x} \in \mathbb{R}^n$ is its reconstruction.

The training objective combines reconstruction error with Frobenius norm regularization:

$$\mathcal{L} = \|W_d W_e x - x\|_2^2 + \lambda \|W\|_F^2 \tag{7}$$

where $\lambda$ is the regularization coefficient (weight decay) and $\|W\|_F = \sqrt{\sum_{i,j} W_{ij}^2}$ is the Frobenius norm of the encoder and decoder matrix.

The reconstruction error term provides the fundamental learning signal for dimensionality reduction. Linear autoencoders optimizing only reconstruction error span a solution space that includes PCA-like solutions (Baldi & Hornik (1989); Plaut (2018)). This connection suggests that the reconstruction objective naturally encourages preservation of data variance structure, which is beneficial for maintaining neighborhood relationships. However, PCA obtains orthogonal basis vectors by constraining the covariance between variables across different dimensions to zero for mapping high-dimensional vectors, which does not necessarily guarantee optimal performance for the k-NN preservation task. A set of non-orthogonal basis vectors may be more conducive to preserving k-NN structure in low-dimensional space, considering that certain directions may convey more information than others. Therefore, our approach extends beyond classical method by incorporating explicit regularization to control transformation properties, seeking potentially better transformations.

The Frobenius norm regularization term serves a crucial role in controlling the singular value spectrum of $W_e$. Mathematically, the relationship between matrix norms provides the key insight:

$$\sigma_{\max}(W_e) = \|W_e\|_2 \leq \|W_e\|_F \tag{8}$$

where $\sigma_{\max}(W_e)$ is the largest singular value and $\|W_e\|_2$ is the spectral norm. By penalizing $\|W_e\|_F$, we constrain the spectral norm and thus the maximum singular value.

The regularization coefficient $\lambda$ directly influences the spectral properties of the encoder matrix. Larger values of $\lambda$ encourage smaller Frobenius norms, which tends to compress the singular value spectrum. Here we tentatively assume that this spectral control mechanism is the key to preserving k-NN structure during dimensionality reduction.

In summary, the optimal value of $\lambda$ achieves a balance: it must be large enough to effectively guide the gradient direction for smaller matrix parameters and more constrained spectral norm, but not so large that it overly constrains the transformation's expressive power, causing the matrix become a sparse representation. In the following section, we provide theoretical analysis to formalize the argument above, establishing the precise mathematical relationship between regularization, singular values, and k-NN preservation performance.

## 3.3 THEORETICAL FOUNDATION

### 3.3.1 FROM k-NN PRESERVATION TO NORM-BOUNDED TRANSFORMATION

To facilitate theoretical analysis, we henceforth focus on the Euclidean distance metric for the remainder of our derivations[1]. Under the Euclidean metric, the distance between vectors $x_i$ and $x_j$ is given by $d(x_i, x_j) = \|x_i - x_j\|_2$. This formulation naturally suggests restricting our attention to linear transformations, which preserve the vector space structure and admit tractable analysis. We therefore consider mapping functions of the form $f(x) = Wx$ where $W \in \mathbb{R}^{m \times n}$ is the transformation matrix.

---

[1] For simplicity, we assume that a dimensionality reduction method preserving Euclidean metric also approximately preserves cosine metric, considering that modern embedding models generate vectors with comparable magnitudes.

Under this framework, the k-NN preservation task can be reformulated as follows: Given $\|x_a - x_i\|_2 \leq \|x_a - x_j\|_2$, where the definition of $x_a, x_i, x_j$ are described in Definition 1, we seek a transformation $W$ such that[2]:

$$\|Wx_a - Wx_i\|_2 \leq \|Wx_a - Wx_j\|_2 \tag{9}$$

By the linearity of $W$, this is equivalent to requiring:

$$\|W(x_a - x_i)\|_2 \leq \|W(x_a - x_j)\|_2 \tag{10}$$

A crucial observation is that since the difference between any two vectors is still a vector in the same space, the k-NN preservation task fundamentally reduces to maintaining the relative ordering of vector norms under transformation. This insight leads us to consider the norm-preserving properties of $W$.

Here we have a natural intuition: a sufficient condition for preserving this norm ordering is to ensure that the transformation preserves norms exactly or scales them uniformly. Specifically, if $W$ satisfies $\|Wv\|_2 = c\|v\|_2$ for all $v \in \mathbb{R}^n$ and some constant $c > 0$, then all distance relationships are preserved perfectly. However, this isometric constraint is overly restrictive for practical dimensionality reduction where $m < n$, as it would require $W$ to be an orthogonal projection scaled by $c$, which is generally impossible when reducing dimensions significantly.

We therefore relax this condition to allow bounded distortion. Define $\delta_i = x_a - x_i$ as the displacement vector between the anchor and its neighbor. We seek transformations that satisfy:

$$(c\varphi_1)\|\delta_i\|_2 \leq \|W\delta_i\|_2 \leq (c\varphi_2)\|\delta_i\|_2 \tag{11}$$

where $c > 0$ represents the nominal scaling factor and $\varphi_2 \geq \varphi_1 \geq 0$ controls the permissible distortion. When $\frac{\varphi_2}{\varphi_1} \to 1$, we recover the strict norm preservation condition. If $\frac{\varphi_2}{\varphi_1}$ is sufficiently close to 1, a transformation satisfying this bounded distortion condition will preserve the k-NN structure with high probability. In other words, $\frac{\varphi_2}{\varphi_1}$ provides a preliminary formal characterization of the upper bound for the rate of norm change.

This relaxation enables us to characterize a broader class of transformations while maintaining theoretical guarantees on neighborhood preservation. The key insight is that by controlling the distortion bound through the spectral properties of $W$, we can achieve robust k-NN preservation even under aggressive dimensionality reduction. The precise relationship between this bound and the singular values of $W$ will be established in the following section.

### 3.3.2 RAYLEIGH QUOTIENT PROPERTIES FOR SINGULAR VALUE-RELATED UPPER BOUND

For a real symmetric matrix $M$ and a non-zero vector $x \in \mathbb{R}^n$, the Rayleigh quotient is defined as:

$$R(M, x) = \frac{x^T M x}{x^T x} \tag{12}$$

A crucial property of the Rayleigh quotient is that it is bounded by the extreme eigenvalues of $M$. Specifically, for any non-zero vector $x$:

$$\lambda_{\min} \leq R(M, x) \leq \lambda_{\max} \tag{13}$$

where $\lambda_{\min}$ and $\lambda_{\max}$ are the smallest and largest eigenvalues of $M$, respectively. The bounds are reached when $x$ is the corresponding eigenvector. We provide a detailed proof of this fundamental result in Appendix A.

For our dimensionality reduction problem, we work with a transformation matrix $W \in \mathbb{R}^{m \times n}$ where $m < n$. Since $W$ is not square, we can leverage the relationship between singular values of $W$ and eigenvalues of the symmetric matrix $W^T W$. Specifically, if $\sigma_1, \sigma_2, \ldots, \sigma_m$ are the singular values of $W$, then $\sigma_i^2$ are the non-zero eigenvalues of both $W^T W$ and $WW^T$.

Consider the norm of the transformed vector $Wx$. $\|Wx\|_2^2$ can be expressed using the Rayleigh quotient of the symmetric positive semi-definite matrix $W^T W$:

$$\|Wx\|_2^2 = x^T W^T W x = \|x\|_2^2 \cdot R(W^T W, x) \tag{14}$$

---

[2]Since bias terms are counteracted in the calculation, we neglect it here.

Applying the bounds of the Rayleigh quotient and taking square roots:

$$\sigma_{\min}\|x\|_2 \leq \|Wx\|_2 \leq \sigma_{\max}\|x\|_2 \tag{15}$$

The connection to our bounded norm transformation requirement(Equation 11) becomes clear. With a clear singular bound for norm change after transformation, for two displacement vectors with the same norm($\|\delta_1\|_2 = \|\delta_2\|_2$), the upper bound of norm distortion between these two vectors is:

$$\frac{\|W\delta_1\|_2}{\|W\delta_2\|_2} \leq \frac{\sigma_{\max}\|\delta_1\|_2}{\sigma_{\min}\|\delta_2\|_2} = \frac{\sigma_{\max}}{\sigma_{\min}} = \kappa(W) \tag{16}$$

where $\kappa(W) = \frac{\sigma_{\max}}{\sigma_{\min}}$ is the condition number of $W$. This establishes that controlling the condition number of $W$ directly enables bounded norm transformations, thus influencing the performance of k-NN preservation. As we discussed in the RAE framework, the Frobenius norm regularization in our loss function provides an effective mechanism for this control over the singular value spectrum, with the regularization coefficient $\lambda$ serving as the key parameter for tuning the condition number.

## 4 EXPERIMENTS

### 4.1 EXPERIMENTAL SETUP

*Datasets and Embeddings.* We evaluated our method on four diverse datasets covering different modalities and embedding dimensions to ensure comprehensive performance assessment. The CelebA dataset (Liu et al. (2015)) was encoded using a pre-trained ViT model (Dosovitskiy et al. (2020)) to 512-dimensional vectors, with 10,000 samples selected for experiments. The IMDb dataset (Maas et al. (2011)) was embedded using MPNet (Song et al. (2020)) to 768 dimensions, selecting 20,000 samples. ImageNet-Tiny (Wu et al. (2017)) was processed through DINOv2 (Oquab et al. (2023)) to generate 384-dimensional features, using 15,000 samples. The Flickr30k dataset (Young et al. (2014)) was encoded using CLIP (Radford et al. (2021)), with image and text embeddings concatenated to form 1,024-dimensional vectors, selecting 15,000 samples. All datasets were split 9:1 for training and testing, with k-NN preservation accuracy measured using the $P_{overall}$ metric from Equation 4.

*Baseline Methods.* We compared RAE against four established dimensionality reduction techniques: UMAP (McInnes & Healy (2018)), Isomap (Tenenbaum et al. (2000)), PCA (Wold et al. (1987)), and MDS with linear extension (Chen et al. (2015); Trosset & Priebe (2008)). Since MDS does not inherently support out-of-sample dimensionality reduction, we first applied MDS to the training set and subsequently learned the projection mapping using a linear regression model for test set transformation.

*Implementation Details.* Experiments were conducted using Python 3.11. RAE training was performed on an NVIDIA GeForce RTX 4060 GPU (8GB memory), while baseline methods and k-NN computations ran on an Intel Core i7-14650HX CPU with 16GB RAM. We utilized the FAISS library (Douze et al. (2024)) for efficient nearest neighbor search. For RAE training, we employed the Adam optimizer with weight decay as the regularization coefficient $\lambda$. Training consisted of 3,000 update steps with a batch size of 128, using cosine annealing to adjust the learning rate from 1e-3 to 1e-5.

### 4.2 K-NN PRESERVATION ACCURACY EVALUATION

We first evaluate the k-NN preservation accuracy $P_{overall}$ of RAE against baseline methods across all datasets when reducing to various target dimensions. Top-5 accuracy across datasets and methods is demonstrated in Table 1.

Overall, PCA and RAE consistently achieve superior k-NN preservation performance, while MDS, ISOMAP, and UMAP struggle to maintain neighborhood relationships after dimensionality reduction. On the ImageNet dataset, RAE achieves the highest accuracy among all methods for both Euclidean and cosine distance metrics across all target dimensions, slightly outperforming PCA. For the remaining three datasets, RAE demonstrates particularly strong performance on the cosine similarity metric, surpassing PCA by at least 12% across all target dimensions. Under the Euclidean

Table 1: Top-5 accuracy across different datasets and methods

| Dataset | Method | Euclidean | Cosine | Euclidean | Cosine | Euclidean | Cosine |
|---|---|---|---|---|---|---|---|
| | | **Dim=128** | | **Dim=192** | | **Dim=256** | |
| | MDS | 57.85 | 58.17 | 63.13 | 63.21 | 51.67 | 56.31 |
| | ISOMAP | 39.41 | 35.47 | 40.24 | 36.15 | 40.52 | 36.57 |
| **ImageNet**(384d) | UMAP | 25.92 | 25.39 | 25.81 | 25.16 | 26.21 | 25.21 |
| | PCA | 70.76 | 74.27 | 81.19 | 82.76 | 88.21 | 88.69 |
| | **RAE** | **71.23** | **74.44** | **81.48** | **83.85** | **88.65** | **89.84** |
| | | **Dim=128** | | **Dim=256** | | **Dim=384** | |
| | MDS | 54.06 | 52.82 | 56.92 | 55.40 | 58.12 | 55.90 |
| | ISOMAP | 35.96 | 34.00 | 38.62 | 36.70 | 39.66 | 37.68 |
| **CelebA**(512d) | UMAP | 21.86 | 20.88 | 21.96 | 20.74 | 21.88 | 21.10 |
| | PCA | **86.10** | 73.78 | **94.94** | 76.28 | **98.20** | 76.68 |
| | **RAE** | 84.38 | **85.12** | 92.32 | **90.26** | 94.20 | **91.82** |
| | | **Dim=256** | | **Dim=384** | | **Dim=512** | |
| | MDS | 44.00 | 43.20 | 45.15 | 43.73 | 45.48 | 44.35 |
| | ISOMAP | 39.92 | 36.61 | 41.29 | 38.26 | 42.07 | 38.99 |
| **IMDb**(768d) | UMAP | 20.63 | 20.89 | 20.85 | 20.82 | 20.65 | 21.02 |
| | PCA | **93.38** | 70.86 | **97.75** | 71.10 | **99.08** | 71.17 |
| | **RAE** | 91.88 | **87.80** | 94.08 | **89.40** | 94.56 | **87.64** |
| | | **Dim=256** | | **Dim=512** | | **Dim=768** | |
| | MDS | 47.44 | 44.40 | 49.95 | 46.64 | 50.00 | 46.77 |
| | ISOMAP | 39.12 | 36.08 | 40.99 | 37.80 | 41.27 | 38.13 |
| **flickr30k**(1024d) | UMAP | 22.08 | 22.15 | 21.80 | 21.85 | 21.80 | 21.49 |
| | PCA | 81.16 | 67.12 | **92.43** | 69.69 | **97.60** | 69.80 |
| | **RAE** | **81.48** | **79.27** | 89.77 | **85.17** | 92.47 | **86.49** |

metric, RAE achieves accuracy comparable to PCA while maintaining high absolute performance. These results demonstrate that RAE effectively preserves k-NN relationships across diverse modalities, embedding models, and distance metrics, establishing its superiority over classical dimensionality reduction methods.

### 4.3 EFFECT OF REGULARIZATION ON CONDITION NUMBER

To validate our theoretical framework connecting regularization to k-NN preservation through condition number control, we analyze how the regularization coefficient $\lambda$ (weight decay) affects the encoder matrix properties and resulting accuracy. We present results showing the relationship between weight decay values and three key metrics: k-NN preservation accuracy, maximum/minimum singular values, and condition number $\kappa(W)$. Result can be seen in Figure 1.

Our experiments reveal that optimal regularization improves k-NN accuracy by at least 8% compared to the unregularized Auto-Encoder. As $\lambda$ increases from small values, the maximum and minimum singular values converge, leading to a decreased condition number and peak k-NN accuracy. Further increasing $\lambda$ beyond the optimal point causes dramatic changes in the singular value spectrum—either the maximum or minimum singular value shifts significantly, rapidly increasing the condition number. This deterioration in conditioning coincides precisely with a sharp drop in k-NN accuracy. These behaviors directly validate our theoretical prediction that controlling the condition number through Frobenius norm regularization enhances neighborhood preservation.

Table 2: Average training and inference time (seconds) across all datasets, measured at target dimension = 50% of original dimension.

| Method | Training | Inference |
|---|---|---|
| PCA | 0.41 | $\approx 10^{-3}$ |
| RAE | 8.18 | $\approx 10^{-3}$ |
| UMAP | 76.78 | 4.75 |
| Isomap | 141.58 | 1.25 |
| MDS | $\gg 541.12^{*}$ | – |

$^{*}$Training on 5000 samples only due to $O(N^3)$ complexity.

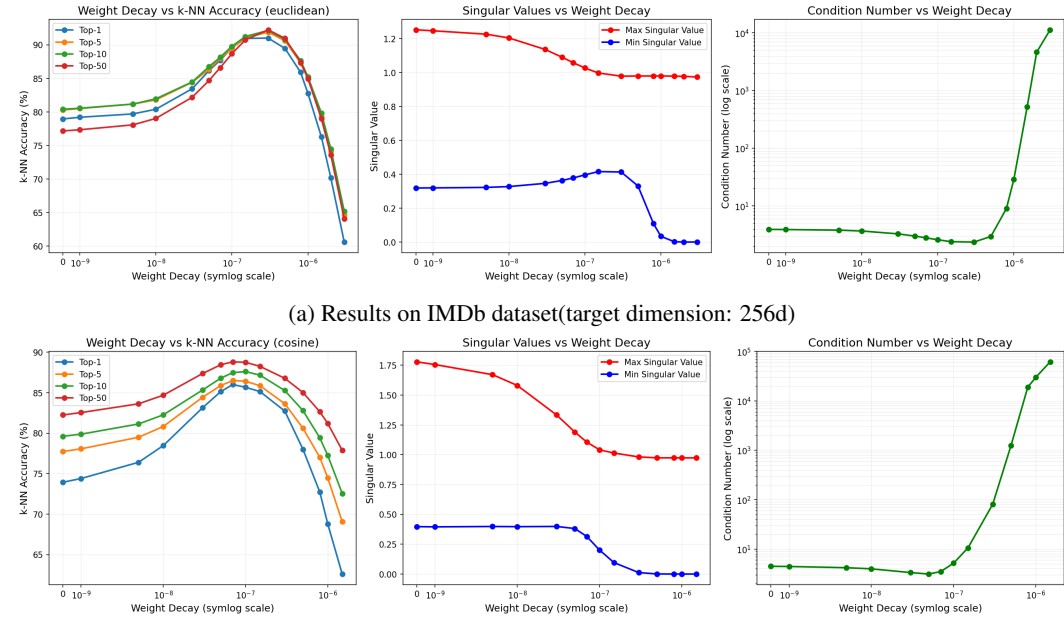

(a) Results on IMDb dataset(target dimension: 256d)

(b) Results on flickr30k dataset(target dimension: 768d)

Figure 1: Weight decay analysis on IMDb (Euclidean, 256d) and Flickr30k (cosine, 768d) datasets. Each row shows k-NN accuracy for different k values, singular value spectrum ($\sigma_{max}$ and $\sigma_{min}$), and condition number $\kappa(W)$ as functions of weight decay $\lambda$. The consistent pattern across datasets—optimal accuracy at minimal condition number—demonstrates the robustness of our regularization strategy.

### 4.4 COMPUTATIONAL EFFICIENCY

Table 2 presents the average training and inference times across all four datasets when reducing to 50% of the original embedding dimension. RAE demonstrates excellent computational efficiency, requiring only 8.18 seconds for training while maintaining inference speed comparable to PCA at the millisecond level. This represents a favorable trade-off considering RAE's significant accuracy improvements. In contrast, other methods incur substantially higher computational costs. UMAP and ISOMAP require 76.78s and 141.58s for training respectively, with inference times in the multi-second range. MDS exhibits prohibitive training complexity due to its $O(N^3)$ scaling, requiring over 541 seconds even when limited to 5000 training samples.

These results establish RAE as practically deployable for large-scale applications. The millisecond-level inference enables real-time use, while the simple two-layer architecture ensures minimal hardware requirements and broad accessibility. This efficiency-performance balance validates RAE as a superior alternative to existing dimensionality reduction methods.

## 5 CONCLUSION

In this paper, we presented RAE, a principled dimensionality reduction framework specifically designed to preserve k-NN structure in high-dimensional embeddings. By incorporating Frobenius norm regularization into the Auto-Encoder framework, we established a direct connection between the regularization coefficient and k-NN preservation accuracy. Another key contribution of RAE lies in that the condition number of transformation matrix provides a well-defined upper bound guarantee of norm distortion rate through the bounding effect of Rayleigh quotient.

Future work will explore alternative regularization formulations beyond the Frobenius norm, potentially uncovering more effective constraints for k-NN preservation. As embedding-based applications continue to proliferate, RAE offers a promising direction for efficient and accurate large-scale vector retrieval.

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

## A    PROOF OF RAYLEIGH QUOTIENT BOUNDS

We prove that for a real symmetric matrix $M$ with eigenvalues $\lambda_1 \leq \lambda_2 \leq \cdots \leq \lambda_n$ and corresponding orthonormal eigenvectors $v_1, v_2, \ldots, v_n$:

$$\lambda_1 \leq R(M, x) \leq \lambda_n \quad \text{for all } x \neq 0$$

**Proof:** Since the eigenvectors form an orthonormal basis, any vector $x$ can be expressed as:

$$x = \sum_{i=1}^{n} y_i v_i \tag{17}$$

where $y_i = x^T v_i$. Then:

$$Mx = \sum_{i=1}^{n} y_i M v_i = \sum_{i=1}^{n} \lambda_i y_i v_i \tag{18}$$

The Rayleigh quotient becomes:

$$R(M, x) = \frac{x^T M x}{x^T x} = \frac{\sum_{i=1}^{n} \lambda_i y_i^2}{\sum_{i=1}^{n} y_i^2} \tag{19}$$

This is a weighted average of the eigenvalues, where the weights $w_i = y_i^2 / \sum_{j=1}^{n} y_j^2$ satisfy $w_i \geq 0$ and $\sum_{i=1}^{n} w_i = 1$. Since any weighted average of values lies between their minimum and maximum:

$$\lambda_1 = \lambda_{\min} \leq \sum_{i=1}^{n} w_i \lambda_i \leq \lambda_{\max} = \lambda_n \tag{20}$$

Therefore:

$$\lambda_{\min} \leq R(M, x) \leq \lambda_{\max} \tag{21}$$

The lower bound is attained when $x = v_1$ (giving $R(M, v_1) = \lambda_1$), and the upper bound when $x = v_n$ (giving $R(M, v_n) = \lambda_n$). $\square$

## B    THE USE OF LARGE LANGUAGE MODELS (LLMs)

We declare the use of Large Language Models (LLMs) in preparing this manuscript for writing assistance and literature discovery. The authors take full responsibility for all the contents we write.

Writing Assistance and Polish: In our writing process, we first articulated our own ideas and formulations completely. Subsequently, we consulted LLMs for grammar corrections and stylistic refinements. After receiving LLM suggestions, we carefully reviewed and further edited the text, iterating this process as needed. This assistance was primarily utilized in text-intensive sections such as the Introduction and Related Work. The Methods and Experimental Analysis sections received minimal LLM assistance to preserve the technical precision of our original expressions.

Literature Retrieval and Discovery: For literature search, we initially conducted our own comprehensive review based on research questions. We then consulted LLMs to identify potentially relevant studies, guiding the AI towards specific areas of interest through iterative queries. Upon receiving suggestions, we rigorously verified each reference by: (1) confirming the actual existence of cited papers, and (2) validating that the AI's interpretation accurately reflected the original content. We explicitly avoided directly incorporating AI-generated citations without verification, as we are aware that LLMs can generate plausible but fictitious references when prompted for specific information.

All technical contributions, theoretical derivations, experimental designs, and analytical insights represent our original work. Every fact and citation has been independently verified against primary sources.

