# OpenReview forum: "RAE: A Neural Network Dimensionality Reduction Method for Nearest Neighbors Preservation in Vector Search"
_ICLR.cc/2026/Conference — Submitted to ICLR 2026_

### Official Review · Reviewer_Scie · 2025-10-28

**Soundness:** 2
**Presentation:** 2
**Contribution:** 1
**Rating:** 2
**Confidence:** 4

**Summary:**

This paper seeks to develop a dimensionality reduction method that preserves k-NN relationships while at the same time achieving strong compression for the vectors. They use an autoencoder, ie a reconstruction loss, together with a regularization strategy that favorstransformations that maintain neighborhood structures. The paper further contains a theoretical analysis that gives bounds on how well the reduced vectors maintain neighbor relationships, with the upper bound controlled by coefficient $\lambda$.

**Strengths:**

The paper is well written and has a easy to follow theoretical analysis. The linear encoder trained is some times better for classification than PCA

**Weaknesses:**

1) Experimental validation lacking for a top-tier paper: The datasets tested are generally small scale: The ImageNet-Tiny dataset used is a 10-way classification task. Showing results on more common/larger datasets would strengthen the paper (eg ImageNet-1k, Oxford+Paris datasets with distractors, Google landmarks)

2) The presentation of the method in the introduction is misleading:  To me the learning method does not __explicitly__ preserves k-NN relationships (as the paper claims). The training objective is an autoencoder with Frobernius norm regularization, while from the analysis in sec 3 it is clear that, kNN preservation is very implicit.

3) Discussion and comparisons to closely related works like [A,B] that are utilizing contrastive learning objectives are missing.
[A] Dimensionality reduction by learning an invariant mapping. (CVPR, 2006).
[B] TLDR: Twin Learning for Dimensionality Reduction (TMLR 2022).

4) Gains over PCA are very small, and in fact PCA is better in many cases.

Other notes:
- Definition 1 "k-NN Preservation Task" does not seem to present some task.
- The tables should name ImageNet-Tiny with its full name.

**Questions:**

1. How does the method **explicitly** preserves k-NN relationships, as is claimed in the introduction? Can you explain how "RAE directly targets the preservation of relative distances between vectors and their neighbors through a carefully crafted optimization objective."

2. Can this be generalized to non linear transformations?

---

### Official Review · Reviewer_6rBo · 2025-10-28

**Soundness:** 2
**Presentation:** 3
**Contribution:** 2
**Rating:** 2
**Confidence:** 3

**Summary:**

The paper focuses on the issue of compressing feature dimensions to improve retrieval efficiency and proposes RAE, a neural network-based feature dimensionality reduction method. During the training process, a Frobenius norm is introduced as a regularization constraint to compress features while preserving the KNN relationships in the high-dimensional space, thereby ensuring retrieval accuracy. In the experimental section, RAE is compared with methods such as PCA, and it demonstrates certain advantages in some scenarios.

**Strengths:**

1)The research problem has strong practical application value.
2)The proposed method is supported by certain theoretical proofs.

**Weaknesses:**

1）The advantages of the proposed method are very marginal. According to Table 1, the improvement of RAE over PCA (a classic algorithm from many years ago) is limited. On ImageNet, the improvements are all around 0.5 percentage points, and only in Top-5 accuracy. On CelebA, it performs on par with PCA, while in the Euclidean experiment, it is significantly weaker than PCA. Similar situations are observed on IMDb and Flickr30K. Compared to PCA, RAE also requires tuning hyperparameters, which severely impacts its performance.
2）Using neural networks for feature compression is not a particularly novel idea, but the authors did not compare their work with such existing methods at all. Although the paper provides some explanations, they are insufficient to convince the reviewers.
3）The scale of the datasets used falls short of the currently promoted billion-scale retrieval benchmarks.

**Questions:**

Please refer to the weaknesses section

---

### Official Review · Reviewer_MoJ9 · 2025-11-01

**Soundness:** 1
**Presentation:** 1
**Contribution:** 1
**Rating:** 0
**Confidence:** 5

**Summary:**

The authors propose a dimensionality reduction technique called RAE. RAE is a linear autoencoder with a Frobenius norm regularization.

**Strengths:**

Given the nature of the weaknesses described below, I do not find any clear strengths in the paper. The proposed method has a flaw, that unless properly explained, is fatal in my opinion. Moreover, there are other dimensionality reduction methods beyond PCA that should be included in the study.

**Weaknesses:**

The main equation of the paper, Equation (7), is not properly written. Are the authors constraining W_e, W_d, their product, or something else? The matrix W is never properly defined. I would assume from the following discussion that the authors are regularizing W_e. Moreover, this equation cannot be the full training objective, but that one for a specific vector in the dataset.

The fact that a linear projection into a lower dimensional space has bounded distortions, given by the ratios of the singular values of the matrix is well known in linear algebra and not a new discovery.

Maybe more importantly, the optimal solution to the problem
$$
min_{W_d, W_e} || W_d W_e X - X ||_2^2 ,
$$
where the columns of X are the data vectors, is given by the singular vectors SVD of the X associated with the top singular values of X. Assuming a hard constraint on W_e, we can set up the new problem

$$
min_{W_d, W_e} || W_d W_e X - X ||_2^2  \quad s.t. \quad ||W_e||_F^2 < \epsilon .
$$

The solution to this problem will still be the same one as the above problem, but W_e and W_d would be scaled down and up by a scalar, respectively. Since the hard constraint problem is equivalent to (a properly written version of) Equation (7) for some \lambda, it seems like the authors are re-inventing PCA but with a scalar factor that changes nothing of the overall structure of the problem. From this perspective, it seems like any differences with PCA in the results are just because of optimization used by the authors to compute W_d and W_e. In my mind, unless properly explained, this is a fatal flaw in this work.

There are previous uses of dimensionality reduction for vector search that the authors should acknowledge and probably compare to. I leave some examples for reference below.

- Jegou, H., Douze, M., Schmid, C. and Perez, P. (2010), Aggregating local descriptors into a compact image
representation, in ‘IEEE Conference on Computer Vision and Pattern Recognition’, IEEE, pp. 3304–3311.

- Gong, Y., Lazebnik, S., Gordo, A. and Perronnin, F. (2012), ‘Iterative quantization: A procrustean approach to learning binary codes for large-scale image retrieval’, IEEE transactions on pattern analysis and machine intelligence 35(12), 2916–2929.

- Babenko, A. and Lempitsky, V. (2014b), ‘The inverted multi-index’, IEEE transactions on pattern analysis and machine intelligence 37(6), 1247–1260.

- Wei, B., Guan, T. and Yu, J. (2014), ‘Projected residual vector quantization for ANN search’, IEEE MultiMedia 21(3), 41–51.

- Zhang, H., Tang, B., Hu, W. and Wang, X. (2022), Connecting compression spaces with transformer for approximate nearest neighbor search, in ‘European Conference on Computer Vision’, pp. 515–530.

- Tepper, M., Bhati, I. S., Aguerrebere, C., Hildebrand, M., & Willke, T. L. LeanVec: Searching vectors faster by making them fit. Transactions on Machine Learning Research.

**Questions:**

Please refer to my remarks above.

---

### Official Review · Reviewer_YVfq · 2025-11-01

**Soundness:** 4
**Presentation:** 4
**Contribution:** 2
**Rating:** 2
**Confidence:** 4

**Summary:**

This paper presents a dimensionality reduction technique with the goal of preserving nearest neighbors. Given a set of embedding vectors the goal is to project them to a lower dimensional space such their k-NN neighbors are preserved. This is achieved using a linear projection from an n to m dimensional space, and the linear projection is learned by minimizing a reconstruction loss with a regularization term.

The k-NN neighbors are preserved by virtue of preserving the pairwise distances between the vectors. Experimental results show that RAE performs better than other traditional methods including MDS, ISOMAP, UMAP, and PCA.

**Strengths:**

- The paper is very well presented and the core concepts and ideas are clear. The mathematical details seem to be correct and reasonably detailed.
- Experimental results show that the method is more accurate compared to prior comparable baselines.

**Weaknesses:**

- Linear dimensionality reduction methods are a well studied topic and as such the novelty of this paper is unclear. The method presented here seems to be very similar to the already existing PCA.
- A higher dimensional vector of dimension n is first projected into a low dimensional subspace of dimension m and projected back to dimension n. This process essentially projects the n dimensional vector in a row-rank subspace of dimension n, which is closely related to PCA. PCA is essentially an orthogonal projection to a low rank subspace and is equivalent to minimizing the reconstruction error between the original and projected vectors. The only difference between the proposed method and PCA seems to be the regularization term.
- The utility of the method is unclear. In the case of retrieval and search using neural embeddings, why would one not train low dimensional vectors directly or perhaps use Matryoshka style embeddings which natively provide low dimensional embeddings for various use cases.

**Questions:**

- Without the regularization term, the method seems to be closely related to PCA. With the regularization term is it similar to probabilistic PCA with the regularization term being similar to prior on the weights? It is not exactly the same, but perhaps similar?
- Does minimizing the loss in Eq. 7 without the regularization term, essentially give us PCA?

---

### Meta-Review · Area_Chair_EMAS · 2026-01-04

**Summary:**

The paper proposes a "Regularized Auto-Encoder" (RAE) for dimensionality reduction, aiming to preserve k-NN relationships in vector search by adding a Frobenius norm regularization to a linear autoencoder.

The reviewers reached a unanimous consensus to reject the paper (Scores: 0, 2, 2, 2). The primary reasons for this decision are threefold:

1. **Theoretical Novelty and Validity:** The most critical concern, highlighted extensively by Reviewer MoJ9 and YVfq, is that the method is similar to PCA. Reviewer MoJ9 effectively pointed out a potential "fatal flaw": adding a Frobenius norm regularization merely scales the singular values but does not change the principal subspace found by PCA. Thus, the proposed method is viewed as re-inventing PCA with a scalar factor, rather than a novel contribution.
2. **Marginal Empirical Gains:** Reviewers 6rBo and Scie noted that the experimental improvements over PCA are marginal (often <0.5%) or non-existent. In some cases (e.g., Euclidean experiment), the method performs worse than PCA. The necessity of hyperparameter tuning for RAE, compared to the parameter-free nature of PCA, makes these marginal gains even less attractive.
3. **Insufficient Benchmarking:** Reviewers 6rBo and Scie noted that the evaluation lacks comparisons with modern state-of-the-art methods (e.g., Matryoshka Embeddings, Product Quantization, Contrastive Learning approaches like DrLIM/TLDR) and relies on small-scale datasets (Tiny ImageNet) rather than the billion-scale benchmarks standard in the vector search literature.

Given the fundamental theoretical challenges and weak empirical evidence, the paper is not ready for publication.

**Reviewer Concerns:**

The authors did not submit a rebuttal. Consequently, none of the reviewers' concerns were addressed. All issues raised in the initial reviews remain outstanding.

**Reviewer Scores:**

Since the authors did not provide a rebuttal, all reviewers would undoubtedly maintain their original scores (Reject/Strong Reject).

---

### Decision · Program_Chairs · 2026-01-26

Reject